# Sex Differences in the Relationship between Chronotype and Eating Behaviour: A Focus on Binge Eating and Food Addiction

**DOI:** 10.3390/nu15214580

**Published:** 2023-10-28

**Authors:** Ramona De Amicis, Letizia Galasso, Riccardo Cavallaro, Sara Paola Mambrini, Lucia Castelli, Angela Montaruli, Eliana Roveda, Fabio Esposito, Alessandro Leone, Andrea Foppiani, Alberto Battezzati, Simona Bertoli

**Affiliations:** 1International Center for the Assessment of Nutritional Status and the Development of Dietary Intervention Strategies (ICANS-DIS), Department of Food, Environmental and Nutritional Sciences (DeFENS), University of Milan, 20133 Milan, Italy; riccardo.cavallaro99@gmail.com (R.C.); sara.mambrini@unimi.it (S.P.M.); alessandro.leone1@unimi.it (A.L.); andrea.foppiani@unimi.it (A.F.); alberto.battezzati@unimi.it (A.B.); simona.bertoli@unimi.it (S.B.); 2Lab of Nutrition and Obesity Research, Istituto Auxologico Italiano, IRCCS, 20145 Milan, Italy; 3Department of Biomedical Sciences for Health, University of Milan, Via Giuseppe Colombo 71, 20133 Milan, Italy; letizia.galasso@unimi.it (L.G.); lucia.castelli@unimi.it (L.C.); angela.montaruli@unimi.it (A.M.); eliana.roveda@unimi.it (E.R.); fabio.esposito@unimi.it (F.E.); 4Laboratory of Metabolic Research, S. Giuseppe Hospital, Istituto Auxologico Italiano, IRCCS, 28824 Piancavallo, Italy; 5IRCCS Ospedale Galeazzi-Sant’Ambrogio, Via Cristina Belgioioso 173, 20161 Milan, Italy; 6Clinical Nutrition Unit, Department of Endocrine and Metabolic Medicine, Istituto Auxologico Italiano, IRCCS, 20133 Milan, Italy

**Keywords:** chronotype, chrononutrition, binge eating, food addiction, sex differences

## Abstract

Background: Men are more likely than women to have subthreshold overeating disorders. Lifestyle plays a role as a determinant, while chronotype is an emerging factor. Chronotype explains the natural preferences of wakefulness and activity throughout the day: evening chronotypes (E-Types), those most productive in the evening, have been linked with unhealthy dietary patterns and a higher propensity to substance addiction than morning types (M-Types). Methods: We carried out a cross-sectional study on 750 overweight or obese adults (70% females, 48 ± 10 years, BMI 31.7 ± 5.8 kg/m^2^). The Binge-Eating Scale, the Yale Food Addiction Scale 2.0 (YFAS 2.0), the reduced Morningness-Eveningness Questionnaire (rMEQ), and the MEDAS questionnaire were used to assess binge eating, food addiction, chronotype, and adherence to the Mediterranean diet, respectively. Results: No differences in BES binge-eating and FA food-addiction scores occurred between chronotypes, but we found significant interactions between sex × rMEQ score. While women showed the same prevalence for binge eating and food addiction across all chronotypes, binge eating and food addiction risk increased with reducing rMEQ score in men, indicating that being male and E-Type increases the risk association of binge eating and/or food addiction prevalence. Conclusions: chronotype is associated with binge eating and food addiction in men, emphasizing the link between chronobiology and sex differences as determinants in appetite and eating behaviour dysregulation and in overweight and obesity.

## 1. Introduction

Overeating is a psychological and behavioural factor contributing to overnutrition, one of the key elements in the manifestation of obesity [1,2]. Depending on the level, it could result in addiction, such as food addiction, or compulsive behaviour, such as binge eating [3].

Food addiction is a novel construct that could lead to the development of different phenotypes of obesity and eating disorders. The core concept revolves around the aspect of certain foods (mostly highly palatable and ultra-high processed foods) being able to promote addictive-like substance abuse behaviour in some individuals [4]. This kind of food shows similarities between sugar, any palatable food, and addictive drugs such as cocaine, amphetamines, nicotine, and morphine [5,6]. The greater availability and related exposure to advertising of low nutritional quality and energy-dense foods that look attractive, hyper-palatable, cheap, and ready to eat [7] lead to higher trends of food addiction, especially in overweight and obese subjects. These circumstances confirm a solid correlation between food addiction and a higher BMI [8]. In this context, binge eating indicates overeating in combination with a lack of control. The latter is defined as eating a quantity of food in a discrete period of time that is unquestionably greater than what most people would consume under similar circumstances [9]. When it occurs at least once a week for 3 months without consistent compensatory behaviours to limit weight gain, unlike bulimia nervosa, it is defined as binge eating disorder [3]. BE is also present in anorexia nervosa, bulimia nervosa, and night-eating syndrome [3], and it is particularly prevalent in individuals who underwent weight-loss treatments, showing increased difficulty in weight maintenance compared to controls and, thus, promoting weight-cycling [10].

Predictors were largely similar across overeating categories: both food addiction and binge eating are predicted by similar risk factors that vary in severity [2], such as female sex, young age, being overweight, sedentariness, compulsive alcohol consumption, insomnia, impulsivity, mood, anxiety disorders, craving and emotional eating [11,12], with chronotype emerging as an influencing factor.

Chronotype represents the individual circadian typology that captures the natural inclination of psychological, behavioural, and biological manifestations with an approximately 24 h oscillation, called a circadian rhythm [13]. Chronotype ranges from morningness to eveningness, with morning-types (M-Types) having their activity peak performance in the first part of the day, while evening-types (E-Types) need time to feel rested after awakening and have their peak level of physical and mental efficiency from the late afternoon [14]. Intermediate or neither-types (N-Types) are individuals without a pronounced circadian preference. E-Types are mostly prime-aged and male adults with unhealthy lifestyles, understood as patterns of physical activity, diet, and sleep hygiene [15,16,17]: they have sedentary activity, smoking habits, and poorer sleep patterns, e.g., fewer sleeping hours, irregular sleep schedules and/or sleep apnoea, than M-Types [18,19,20]. Sex differences in chronotyping are also reported, with most studies showing that males tend to be more likely than females to be in the evening/late chronotypes [19]. Concerning eating habits, E-Types are characterized by lower adherence to a healthy diet, such as the Mediterranean diet: they consume more calories in terms of proteins, simple carbohydrates, and saturated fatty acids during the evening. These calories mainly come from confectionery food, fast foods, and sugar-sweetened beverages, resulting in an overall lower intake of minerals and vitamins from fruits, vegetables, fish, pulses, and cereals [19,20,21]. Their eating behaviour is typified by night times, larger food portion sizes, and longer time intervals between meals. Moreover, they show a greater propensity for food cravings, emotional eating, reduction of control over diet, and the problem of controlling the amount of food eaten [20,22], probably caused by a higher prevalence of impulsivity, anxiety, and depression [23]. All these aspects lead to an altered body composition, such as higher body mass index (BMI), waist circumference and body fat, and the consequent higher prevalence of metabolic disorders [24]. Accumulating evidence proposes the disruption of the light–dark cycle, typical of E-Types, as a risk factor for obesity disease, food/eating addiction, and substance dependence, which risks triggering the onset of eating disorders [11,25]. In line with what has just been stated, food addiction is associated with insomnia and impulsivity [11], and binge eaters usually have substandard sleep quality and sleepiness during the day [26]. A recent narrative review highlighted how E-Types are more prone to suffer from an eating disorder such as food addiction or night-eating syndrome in every age group [27], especially if they are males. However, the studies focused on healthy subjects and did not include known factors influencing the risk of developing binge eating or food addiction (adherence to the Mediterranean diet, physical activity, smoking, nutritional status) as potential variables influencing the link between eveningness and the development of eating disorders. Indeed, overweight and obesity are associated or may be the manifestation of altered eating behaviour, with known differences in the pathogenesis between males and females [28,29]

Despite these sex differences in both eating behaviour and chronotype, to the best of our current knowledge, no previous study has investigated whether and how the interaction between sex and chronotype may influence the development of various pathologies as alterations in eating behaviour [19]. In such a scenario, we carried out a cross-sectional study to investigate the sex differences in the relationship between chronotype and either binge-eating disorder or food addiction in Caucasian adults with overweight or obesity, considering lifestyle factors.

## 2. Materials and Methods

### 2.1. Study Design

We consecutively recruited Caucasian participants with overweight and obesity spontaneously attending the International Centre for the Assessment of Nutritional Status and the Development of Dietary Intervention Strategies (ICANS-DIS), Department of Food, Environmental and Nutritional Sciences (DeFENS), University of Milan, and the Istituto Auxologico Italiano Piancavallo. They are two Italian clinical research centres specialized in managing subjects with different levels of obesity and its comorbidities to undergo a structured nutritional evaluation. The inclusion criteria were being of adult age (≥18 years), body mass index (BMI) > 25 kg/m^2^, and Caucasian race. Exclusion criteria were (1) pregnancy and nursing; (2) diagnosed cardiovascular, neurological, endocrine, and major psychiatric disorders (e.g., schizophrenia, major depression, history of psychiatric disorders); (3) diagnosis of obstructive sleep apnoea; and (4) any use of medication affecting appetite, sleep, and eating behaviour.

Each subject underwent a clinical and anthropometric evaluation and filled out validated questionnaires to assess their dietary pattern and eating behaviour.

The study complied with all tenets of the Declaration of Helsinki, and each involved participant signed written informed consent. The study procedures were approved by the Ethical Committee of the University of Milan (n. 6/2019).

### 2.2. Clinical and Anthropometrical Assessment

A detailed medical interview was conducted, and self-reported diagnosis of medical conditions and current drug therapies were collected.

Smoking and physical activity were assessed by asking participants as follows: “Do you smoke?”, “If yes, how many cigarettes per day?”, Have you ever smoked before?”, “Do you engage in any kind of structured physical activity?” and “How many hours per week do you spend doing this kind of activity?”. Participants who currently smoke or smoked at least 1 cigarette per day for at least 3 consecutive months were considered smokers or ex-smokers, respectively. Participants who spent 2 h per week in any type of structured exercise were considered active.

Anthropometric measurements were collected according to the conventional criteria and measuring protocols elaborated by Lohman et al. [30]. Body weight (kg) and body height (cm) were taken to the nearest 100 g and 0.1 cm, respectively. Body weight was measured on a column scale (Seca 700 balance, Seca Corporation) with participants wearing only light underwear and after bladder emptying. Body height was measured using a vertical stadiometer. The body mass index (BMI) was calculated using the following formula: body weight (kg)/body height^2^ (m^2^). Obesity was classified according to the WHO guidelines.

### 2.3. Eating Behaviour

To determine binge-eating behaviour, the validated Binge-Eating Scale was used. This 16-item scale assesses a variety of dietary habits, including but not limited to mood, cognition, and behaviour. Additionally, it evaluates the influence of environmental factors on weight control, including media pressure, social isolation, and daily self-weighing. Binge eating is defined as a Binge-eating score ≥ 18 [31].

The Yale Food Addiction Scale 2.0 (YFAS 2.0) is a validated 35-item survey with an eight-point scoring system [32]. According to food and eating, the YFAS 2.0 assesses the eleven substance-related and addictive disorder criteria. At least two of the eleven criteria for substance-related and addictive disorders must be met, and there must be severe impairment for the prospective “diagnosis” of a food addiction. Scores range from 0 to 11, with the following cut-offs: mild = 2–3 symptoms plus impairment or suffering; moderate = 4–5 symptoms plus impairment or suffering; severe = 6 or more symptoms plus impairment or suffering.

### 2.4. Chronotype

Chronotype was measured using an abbreviated 5-item version of the standard 19-item Morningness-Eveningness Questionnaire (MEQ). The MEQ is one of the most widely used chronotype questionnaires due to its good stability, reliability, coefficient range, and translation and validation in several languages. There is 83% of the total variance of the original MEQ accounted for by this shortened version (rMEQ). These five items measure sleep and awake times, peak times, morning wakefulness, and self-perceived chronotype. This value varies from <12 (extreme E-Types, more active later in day) and >17 (extreme M-Types, feeling alert and fresh at dawn). Intermediate scores were associated with N-Types (12–17 points), who have a more flexible sleep period. The total scores were analysed either as a continuous variable or as a categorical variable, with the scores divided in tertiles. The bottom tertile was used to represent eveningness, the middle tertile was used to represent intermediate preference, and the top tertile was used to represent morning preference [26,29].

### 2.5. Adherence to the Mediterranean Diet

A validated 14-item questionnaire was used to assess adherence to the traditional Mediterranean diet. The guidelines of the Prevencion con Dieta Mediterranea (PREDIMED) study group (www.predimed.es, accessed on 10 January 2019) were used to obtain the Mediterranean Adherence Score (MEDAS). These guidelines were adapted from those used in previous studies [18,24,25,26,27]. A score of one point was assigned to each of the following items: (1) olive oil as main cooking fat; (2) olive oil ≥ 4 tablespoons/d; (3) vegetables > 2 servings/d (or ≥ 1 portion raw or salad); (4) fruit ≥ 3 servings/d; (5) red or processed meat < 1 serving/d; (6) butter or cream or margarine < 1/d; (7) sugar-sweetened beverages < 1/d; (8) wine ≥ 3 glasses/wk; (9) legumes ≥ 3 servings/wk; (10) fish/seafood ≥ 3 servings/wk; (11) commercial candy and sweets < 3/wk; (12) nuts ≥ 1/wk; (13) white meat more than red meat (yes); and (14) use of sofrito ≥ 2/wk. Participants scoring ≥ 9 on the MD were considered having a dietary pattern according to the Mediterranean diet [28,30,31].

### 2.6. Statistical Analysis

All continuous variables are presented as mean ± standard deviation (SD) as they followed a Gaussian distribution. Frequencies and proportions are reported for discrete variables. One-way ANOVA with Bonferroni post hoc test for numerical variables was used to compare descriptive characteristics between chronotype groups. Categorical variables were compared using the Χ2 test.

We assessed the association of rMEQ score with the continuous outcomes of interest (BES score and YFAS score) using prespecified linear regression models and robust confidence intervals. We performed 2 regression analyses, either having as a predictor the “binge-eating score” (continuous) or the “food addiction” score (continuous) measured with the “Binge-Eating Scale” and “Yale Food Addiction Scale 2.0” questionnaire, respectively. Both models had the following predictor variables:sex (categorical: 0 = female, 1 = male)age (continuous, years)smoking (categorical: 0 = never smoked, 1 = smoker, ex-smoker)physical activity (categorical: 0 = sedentary, 1 = physically active)body mass index (continuous, kg/m^2^)Mediterranean diet adherence score (continuous)reduced Morningness-Eveningness Questionnaire score (continuous)sex × reduced Morningness-Eveningness Questionnaire score interaction

The variables were selected within the known predictor of either binge eating or food addiction that would explain part of the variance of these outcomes, producing a more precise estimate of the effect of chronotype (dependent on sex) on the outcomes. In order to examine interactions in multiple regression analysis and to exclude multicollinearity, we conducted a test with Variance Inflation Factors (VIFs), and we eventually excluded all variables with VIFs greater than 5 to 10 and lower than 0.1 to 0.2 [33].

The sample size calculation was based on previous studies that investigated the predictors of binge eating or food addiction [34,35]. With 80% power and a 5% significance level, it was estimated that a sample of 243 participants for binge eating and 571 for food addiction was sufficient to test the sex differences in the relationship between chronotype and eating behaviour [36]

Statistical significance was defined as a *p*-value < 0.05. Statistical analysis was carried out with IBM SPSS Statistics software version 28.0 for Windows (IBM, Armonk, NY, USA).

## 3. Results

Of the 856 participants who came to ICANS-DIS, 106 were excluded: 88 (10%) did not meet the inclusion criteria and 18 (2%) did not complete all questionnaires. A total of 750 participants were included in the analysis with an eligibility rate of 88% (see Figure 1).

The general characteristics of the study subjects are reported in Table 1.

A total of 70% (*n* = 525) of participants were women and 30% (*n* = 225) were men. In our population, 32% (*n* = 240) were M-Types, 9% (*n* = 68) were E-Types, and the remaining were N-Types. A total of 21% (*n* = 158) of participants reported food addiction and 13% (*n* = 98) had a probability of binge eating, while only 6% (*n* = 45) reported a risk of having both food addiction and binge eating.

No difference in BMI, binge eating, and food addiction prevalence occurred among the chronotypes. However, rMEQ scores were significantly lower in E-Types compared to M-Types (10 ± 1 vs. 19 ± 1, *p* < 0.001), as expected, and Mediterranean diet adherence was lower in E-Types (6 ± 2 vs. 7 ± 2, *p* < 0.05).

### 3.1. Association between Chronotype and Binge-Eating Behaviour

Younger age and growing BMI resulted as predictors for binge eating (−0.104, 95% CI = −0.17–−0.04, *p* < 0.001; 0.150, 95% CI = 0.04–0.26, *p* = 0.008; respectively). In addition, we found a significant interaction between sex and the chronotype’s score (−0.406; 95% CI = −0.79–−0.002; *p* = 0.037), suggesting a sex difference in the association between binge eating and chronotype, as shown in Table 2.

Figure 2 shows that females tend to have higher binge eating than males regardless of morningness-eveningness categories, while E-Type males show higher levels of binge eating than M-Type males, with levels similar to females.

### 3.2. Association between Chronotype and Food Addiction

No differences in food addiction occurred among chronotypes and both sexes, similar to for binge eating prevalence. However, we found a significant interaction between sex and the chronotype’s score (−0.244, 95% CI = −0.18–−0.31, *p* = 0.039), as shown in Table 3.

Figure 3 shows that females tend to have higher food addiction than males regardless of morningness-eveningness categories, while E-Type males show higher levels of food addiction than M-Type males, with levels similar to females.

## 4. Discussion

We investigated the sex differences in the relationship between chronotype and overeating, such as binge-eating behaviour and food addiction, considering lifestyle factors such as physical activity, smoking habits, and adherence to the Mediterranean diet. Overall, we found that being E-Type was an overeating predictor only in men: notably, E-Type males show higher levels of both binge eating and food addiction than M-Type males, and reach levels similar to those of females, which are higher in all categories of chronotype. In fact, we found no correlations between chronotype and pathological overeating in females at any level, confirming only young age, high BMI, and low adherence to Mediterranean diet as predictors of food addiction. In particular, we found that the direct effect of chronotype on binge-eating and food-addiction scores was not significant, as reported in previous studies [15,17,20,21,22,37], but the interaction between sex and chronotype was significant, highlighting how the effect of chronotype on the development of disorders such as eating disorders is mediated by sex differences. This may be due to the fact that our sample consists of middle-aged participants (30–65 years), in whom sex differences in chronotype expression are still relevant [19], but also to the fact that many aspects of lifestyle, such as smoking, exercise, and dietary patterns, were taken into account when examining this relationship. Future studies would need to be evaluated in the older population, where these differences tend to diminish until they disappear altogether.

The association of the later chronotype (eveningness) and higher scores of binge eating and food addiction are consistent with a recent systematic review on a different population [38] which shows that E-Type children and adolescents had higher BMI, consumed more junk food, and suffered from food addiction and night-eating syndrome. This illustrated how lifestyle factors such as low adherence to the Mediterranean diet can partially explain the contribution of chronotype to compulsive overeating. Little but growing evidence is trying to investigate how circadian rhythm disruptions may characterize either symptoms or modulators of overeating, which could be explained by the link between insomnia and night-eating syndrome [39] but also by the loss of the circadian rhythmicity of metabolic and hedonic regulation of feeding due to the social jet-lag and a gradual increase of prevalence of emotional eating [22,26]. These results are also consistent with the findings of Aoun et al. [23]: they found that healthy male M-Types have lower uncontrolled eating and more cognitive restraint related to food when compared to male E-Types, who are more likely to engage in addictive eating patterns [11]

In addition to the high propensity of E-Types to addiction and substance dependence [40], such as high-calorie foods and beverages, caffeine, alcohol, and fast food, especially at night and during weekends [26], it is well known that binge eaters and subjects with food addiction are more prone to sleep and circadian disturbances. Our results show how these aspects are more probable in men and could explain the link between eveningness, food addiction, and male sex. However, how to explain these sex differences still remains to be investigated. Some biological mechanisms could probably act together. Parikh et al. [41] showed how different responses to solar UV exposure induce food-seeking behaviour in men but not in women, with an increase in ghrelin levels and appetite in men during the daytime; this, when combined with the lack of time due to increasing work schedules, results in overeating in the evening hours or during the weekend. Joye et al. [42] explained how sex hormones and genetic sex regulate nearly all circadian parameters because of sex-specific differences in the cells of the suprachiasmatic nucleus influenced by sex hormones and, at the same time, influence circadian behaviour. In addition, disruptive appetitive behaviour, typical of subjects with binge eating and food addiction later in the daytime, could serve as an external signal to the central circadian pacemaker in maintaining and fuelling the vicious cycle between eveningness and overeating [43].

What can be deduced, however, is that chronobiology and its sex differences need to be involved in lifestyle analysis in overeating disorders, so that dietary and behavioural treatment can be tailored according to chronotype and sex. More specifically, nutritional and behavioural therapies might differ from chronotype to chronotype: e.g., during the COVID-19 lockdown, a well-monitored hypocaloric diet based on the Mediterranean diet model improved binge-eating behaviour and contributed to the individuals’ return to their chronotype and physiological eating rhythms [44]. Moreover, in a U.S. intervention in women with gestational diabetes, a chrononutritional and sleep hygiene intervention significantly reduced the proportion of women with suboptimal glycaemic control thanks to the reduction of carbohydrate intake during the evening hours [45]. Additionally, women with the evening chronotype experienced significant weight loss and a reduction in fat mass, as well as an increase in free-fat mass and phase angle compared with women with the morning chronotype, reporting the chronotype score as the main predictor of weight loss during a dietary intervention with the very low-calorie ketogenic diet [46]. Furthermore, E-Types, when compared with M-Types, lost less weight (percentage of excess weight loss) after bariatric surgery (*p* = 0.015) and, from the fourth year after the bariatric surgery, they reported a higher weight regain (*p* < 0.05), especially in subjects with C-Carriers of the CLOCK gene, a component of the circadian system that displays a less robust circadian rhythm with sleep reduction and increased values of plasma ghrelin [47].

To sum up, the circadian and genetic assessment could provide tailored weight loss recommendations in subjects who underwent bariatric surgery. In this scenario, chronotype could be considered a valuable tool to recognize a particular set of participants needing more intensive treatments. If M-Types are to find beneficial outcomes from the first line of interventions, being male and E-Type, and, therefore, more susceptible to a greater change in body composition and a higher risk of addiction and eating disorders, it would require a more invasive nutritional and behavioural intervention: the amount of calories deficit and the dietary composition must be considered, as well as the timing and frequency of intake combined with surgery and medication. We are in an era where personalized approaches are promising to reduce the burden of several current problems in nutrition research, and the incorporation of Machine Learning into Precision Nutrition research could help us to better phenotype and treat different subjects with different levels of diseases [48].

To the best of our knowledge, this is the first study among adults with overweight and obesity to examine the interaction of sex differences in the contribution of chronotype to overeating, both regarding binge eating and food addiction. Our study contributes to the growing body of evidence on the interaction between chronotype and eating behaviour, and the role of both sex and lifestyle in the interaction. Notwithstanding these critical strengths and considerations, some potential limitations need to be emphasized. The sample size of men was lacking compared to women; however, the dataset contained ~94 observations per degree of freedom of the model used to describe the relationship between chronotype and either binge-eating disorder or food addiction, and so should be able to adequately express the relationship without the risk of overfitting of bias in the interpretation of the results [49]. Additionally, only the risk of dietary change could be analysed, since the results of the questionnaires were not confirmed by expert diagnosis. Finally, the cross-sectional design makes it impossible to establish a cause-and-effect relationship. Therefore, further research is necessary in order to understand this association and prospective longitudinal studies are needed to identify causality between chronotype and eating disorders.

## 5. Conclusions

In conclusion, these results showed that chronotype is associated with binge-eating behaviour and food addiction, but only in men, emphasizing the role of chronobiology and sex differences as determinants of appetite dysregulation in overweight and obesity.

## Figures and Tables

**Figure 1 nutrients-15-04580-f001:**
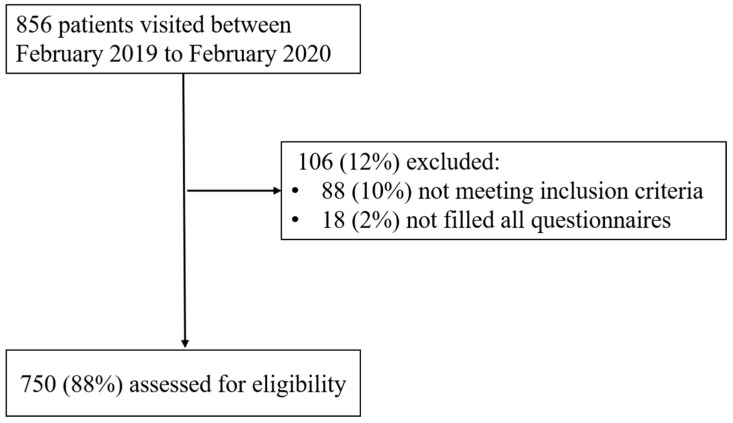
Participant flow diagram.

**Figure 2 nutrients-15-04580-f002:**
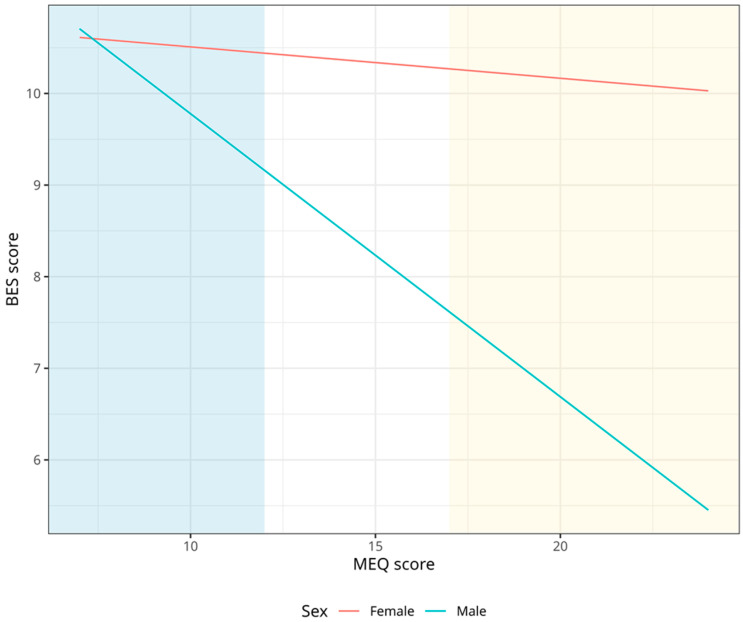
Association between chronotype and binge eating, stratified by sex. Blue represents eveningness and Yellow morningness.

**Figure 3 nutrients-15-04580-f003:**
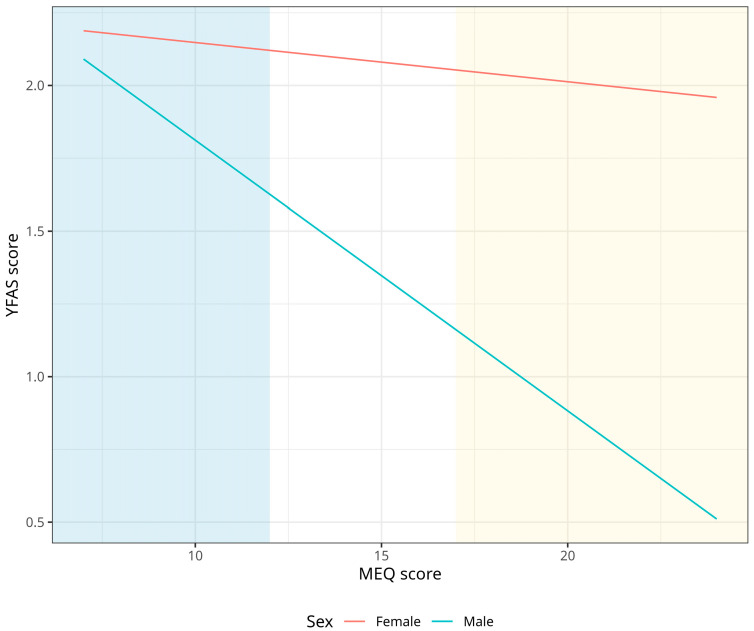
Association between chronotype and food addiction, stratified by sex. Blue represents eveningness and Yellow morningness.

**Table 1 nutrients-15-04580-t001:** General characteristics of the study participants.

	Women (70%, *n* = 525)			Men (30%, *n* = 225)		
	M-Type	N-Type	E-Type	M-Type	N-Type	E-Type
Age (y)	51 ± 11	49 ± 13	47 ± 14	52 ± 12	45 ± 13	46 ± 14
BMI (kg/m^2^)	29.6 ± 6.3	28.9 ± 6.6	28.2 ± 5.9	30.6 ± 5.3	30.4 ± 5.5	30.7 ± 4.4
rMEQ score	19 ± 1	15 ± 2 °	10 ± 1 *	19 ± 1	15 ± 2 °	10 ± 1 *
MEDAS	7 ± 2	7 ± 2	7 ± 2	7 ± 2	7 ± 2	6 ± 2 *
BES score	10 ± 7	10 ± 8	9 ± 6	7 ± 5	8 ± 6	9 ± 6
YFAS score	2 ± 3	2 ± 3	2 ± 3	2 ± 2	2 ± 2	2 ± 2

° *p* < 0.05 vs. M-Types; * *p* < 0.001 vs. M-Types. M-Type = morning, N-Type = neither, E-Type = evening, MEDAS = MEDiterranean Adherence Score, BES = Binge Eating Scale, YFAS = Yale Food Addiction Scale; rMEQ = reduced Morningness-Eveningness Questionnaire.

**Table 2 nutrients-15-04580-t002:** Multivariate linear regression analysis of binge-eating score from the “Binge-Eating Scale” questionnaire.

	β	95% CI	*p*
Sex	0.360	−0.26–0.98	0.254
Age	−0.104	−0.17–−0.04	<0.001
Smoking	0.555	−14.71–25.80	0.591
Physical activity	0.134	−14.31–16.91	0.866
BMI	0.150	0.04–0.26	0.008
MEDAS	−0.142	−0.49–0.20	0.420
rMEQ score	0.125	−0.09–0.34	0.262
Sex × rMEQ	−0.406	−0.79–−0.02	0.037

Dependent variable: BES score; BMI = body mass index; MEDAS = MEDiterranean Adherence Score; rMEQ Score = reduced Morningness-Eveningness Questionnaire; BES Score = Binge-Eating Scale score.

**Table 3 nutrients-15-04580-t003:** Multivariate linear regression analysis of food-addiction score from the “Yale Food Addiction Scale” questionnaire.

	β	95% CI	*p*
Sex	0.001	−0.05–0.05	0.970
Age	−0.015	−0.03–−0.01	0.042
Smoking	0.121	−0.48–0.72	0.691
Physical activity	−0.026	−0.63–0.68	0.933
BMI	0.113	0.09–0.14	0.000
MEDAS	−0.124	−0.23–0.02	0.019
rMEQ score	0.129	−0.12–0.34	0.273
Sex × rMEQ	−0.244	−0.18–−0.31	0.039

Dependent variable: YFAS score; BMI = Body Mass Index; MEDAS = MEDiterranean Adherence Score; rMEQ Score = reduced Morningness-Eveningness Questionnaire; YFAS Score = score of Yale Food Addiction Scale 2.0.

## Data Availability

The data presented in this study are available on request from the corresponding author.

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
