# Peer review of "Sex Differences in the Relationship between Chronotype and Eating Behaviour: A Focus on Binge Eating and Food Addiction"

_nutrients, 2023, doi:10.3390/nu15214580_

Round 1
Reviewer 1 Report
Thank you for the opportunity to review the manuscript. “Gender differences in overeating: does chronotype matter?” is interesting and provides new insights into gender differences in chronotype in overeating. However, I have several important concerns, and thus, I cannot recommend the publication of this manuscript. The greatest concern is the novelty of the study. The authors stated that no studies have examined the association between chronotypes and binge eating (BE) and food addiction (FA). However, BE and FA were mentioned in several papers and review articles on chronotypes (e.g., https://www.frontiersin.org/articles/10.3389/fnins.2022.811771/full). In their discussion, the authors also mentioned papers that have examined the association between chronotype and FA. Therefore, the authors need to state the novelty of this study by citing the appropriate literature.
Next, the authors need to indicate whether the focus of this study is on sex differences, the interaction between sex and chronotype, or both, along with the theoretical basis (depending on the revisions, Tables 2.3 and 4.5 may be used only). The logical development is not consistent from the title to the introduction, methods, results, and discussion.
Finally, the position of the Mediterranean diet in this study is ambiguous. Additionally, throughout the study, the authors should unify the wording "gender" and "sex".
The following suggestions will help the authors reorganize this study.
1. The authors should add references in the following sentences: L. 43-45, 57-59, 145-148.
Title.
2. I understood that BE and FA are the main outcomes of this study. Therefore, I think the title should include BE and FA, not overeating.
Abstract
3. The authors need to state the purpose of this study.
4. Please spell out BES.
Introduction
4. L.64 The authors should specify what the DETERMINANTS are for.
5. L.78-79 Please add what exactly the authors mean by poorer sleeping patterns.
6. The authors need to state in more detail why they focused on Gender differences and the rationale for limiting the subjects to overweight and obese populations.
7. It is not clear from the introduction why Figure 1 was created. Please add a description of the theoretical background, rationale, and purpose of why this analysis was conducted.
Methods
8. L. 102-112 Please describe in detail the process from subject recruitment to sample selection for analysis. Did the authors analyze data from all 750 recruited subjects? How many participants were excluded based on exclusion criteria? I recommend adding a figure of the data collection flow to accurately describe the sample selection process and to facilitate the reader's understanding.
9. L. 108 The authors stated that the only inclusion criterion was age 18 or older, does this not include BMI or race?
10. L. 110-112 I think the exclusion criteria should include the use of sleeping pills.
11. L. 111 There are two 3rd exclusion criteria.
12. L. 130-171 The description of the study items should start with the primary variable of the study. The adjustment variable, MD, should be listed after eating behaviors.
13. Please describe how physical activity and smoking habits were measured.
14. The authors should describe the validity of the sample size. If the sample size was insufficient to perform the analyses chosen by the authors, the robustness of the study method is estimated as low.
15. L.172-185 The analysis section should be detailed so that anyone can replicate the analysis using the authors' data. The authors should report in detail on all analytical methods presented in the Results section. For example, it is difficult to discern what the dependent and independent variables in a multiple regression analysis were, respectively. It is also preferable to perform a single regression analysis. It is difficult to read that a multiple regression analysis that included the entire sample and examined the interaction between sex and chronotype and multiple regression analysis by sex, respectively, were performed. A statement regarding Figure 1 needs to be added. For the multiple regression analysis, it is necessary to specify why the forced entry method was chosen instead of the stepwise method.
Results
16. Please rephrase these sentences in the results section so that the directionality of the results is clear throughout the results. For example, is a "slower (Eveningness)" chronotype associated with an "increase" in BES (more severe BE)?
17. L.190 The term "patients" is confusing and should be unified with "participants".
18. L.200 I think this sentence would be more appropriate to be moved the previous section.
19. If the authors think this analysis of Tables 3 and 5 are necessary, the meaning of these analyses should be described in the analysis section. The interaction between sex and chronotype is not mentioned at all in the Introduction to Methods section.
20. L.233-234 It is difficult to understand from which part of the results this sentence is derived.
Table
21. Tables and figures need to be presented in a style that can be understood independently of the text. Please revise the titles to be more specific and appropriate throughout.
22. Table 1 It is unclear what the p-value represents. The authors need to show the results of the ANOVA and the post-test in Table 1.
23. Table 2 What does "Estimate" mean? From the values, it looks like a beta. Additionally, only the Estimate for Sex has a very different value. Please correct it if necessary.
24. I am not sure what the difference is between Table 2 and Table 4. They have almost the same title, but Table 4 may be the result of FA.
Discussion
25. L.239-246 The authors should also mention the results of Figure 1 in this paragraph.
26. L.247 What "these results" mean ? If all the results are consistent with previous studies, the novelty of this study is lost.
27. L. 312-313 It is unclear from what analysis the authors report that the sample size of men was lacking. Moreover, if the sample size was indeed lacking, the academic value of this study would be greatly compromised.
Author Response
#Reviewer 1
Thank you for the opportunity to review the manuscript. “Gender differences in overeating: does chronotype matter?” is interesting and provides new insights into gender differences in chronotype in overeating. However, I have several important concerns, and thus, I cannot recommend the publication of this manuscript. The greatest concern is the novelty of the study. The authors stated that no studies have examined the association between chronotypes and binge eating (BE) and food addiction (FA). However, BE and FA were mentioned in several papers and review articles on chronotypes (e.g., https://www.frontiersin.org/articles/10.3389/fnins.2022.811771/full). In their discussion, the authors also mentioned papers that have examined the association between chronotype and FA. Therefore, the authors need to state the novelty of this study by citing the appropriate literature.
We thank the reviewer for the opportunity to significantly improve our work.
We have mentioned the individual papers in the review, now adding the narrative review in the introduction as well, which provides a more useful and broader view.
Next, the authors need to indicate whether the focus of this study is on sex differences, the interaction between sex and chronotype, or both, along with the theoretical basis (depending on the revisions, Tables 2.3 and 4.5 may be used only). The logical development is not consistent from the title to the introduction, methods, results, and discussion.
We aimed to describe the relationship between chronotype and either binge eating disorder or food addiction, adjusting for known factors influencing the risk of developing binge eating or food addiction (age,BMI,adherence to mediterranean diet, physical activity level, allowing for a different relationship between sexes, as sexes tend to behave differently in the pathogenesis of these disorders. So we have retained Table 2 and 4 and eliminated table 3 and 5 from the paper.
Finally, the position of the Mediterranean diet in this study is ambiguous.
We have previously studied the relationship between the Mediterranean diet adherence score and either binge eating (doi: 10.1016/j.clnu.2014.02.001.), highlighting that it represents a protective factor. So we added it as a protective lifestyle variable equal to physical activity, smoking, etc.
Additionally, throughout the study, the authors should unify the wording "gender" and "sex".
We thank the reviewer. Fixed it unifying with “sex”.
The following suggestions will help the authors reorganize this study.
- The authors should add references in the following sentences: L. 43-45, 57-59, 145-148.
Fixed it.
Title.
- I understood that BE and FA are the main outcomes of this study. Therefore, I think the title should include BE and FA, not overeating.
Fixed it.
Abstract
- The authors need to state the purpose of this study.
We thank the reviewer for the suggestion. We have rephrased : “A recent narrative review highlights how E-Types are more prone to suffer from an eating disorder such as food addiction or night eating syndrome in every age group, especially if they are males. However, the studies focused on healthy subjects and did not include known factors influencing the risk of developing binge eating or food addiction (adherence to the Mediterranean diet, physical activity, smoking, nutritional status) as variables influencing the link between eveningness and the development of eating disorders.
In such a scenario, we carried out a cross-sectional study to investigate the relationship between chronotype and either binge eating disorder or food addiction in Caucasian adults with overweight or obesity, considering both sex differences and lifestyle factors”.
- Please spell out BES.
Fixed it. We have spelled out most of the abbreviations.
Introduction
- L.64 The authors should specify what the DETERMINANTS are for.
We have rephrased the sentence: “Predictors were largely similar across overeating categories: both food addiction and binge eating are predicted by similar risk factors that vary in severity [11], as female sex, young age, being overweight, sedentariness, compulsive alcohol consumption, insomnia, impulsivity, mood, anxiety disorders, craving and emotional eating [12,13], with chrono-type emerging as influencing factor.”
- L.78-79 Please add what exactly the authors mean by poorer sleeping patterns.
Fixed it: “poorer sleep patterns, eg. less sleeping hours, inconsistent sleep schedules and/or sleep apnea,”
- The authors need to state in more detail why they focused on Gender differences and the rationale for limiting the subjects to overweight and obese populations.
Overweight and obesity are associated or may be the manifestation of altered eating behavior, with known differences in the pathogenesis between men and females. We added this sentence: “the studies focused on healthy subjects and did not include known factors influencing the risk of developing binge eating or food addiction (adherence to the Mediterranean diet, physical activity, smoking, nutritional status) as variables influencing the link between eveningness and the development of eating disorders, as overweight and obesity that are associated or may be the manifestation of altered eating behavior, with known differences in the pathogenesis between men and females (doi: 10.1093/scan/nsz085., 10.1016/j.clinthera.2020.12.003, https://doi.org/10.3390/brainsci12121663).”
- It is not clear from the introduction why Figure 1 was created. Please add a description of the theoretical background, rationale, and purpose of why this analysis was conducted.
We have replaced Figure 1 with 2 figures representing the partial effects of the models shown in 2 and 3 (new numbering), representing the relationship between chronotype and either binge eating or food addiction, stratified by sex. To produce such figures, other predictor variables were kept constant, and in particular for continuous variables the median was chosen, and for categorical variables the most frequent category.
Methods
- L. 102-112 Please describe in detail the process from subject recruitment to sample selection for analysis. Did the authors analyze data from all 750 recruited subjects? How many participants were excluded based on exclusion criteria? I recommend adding a figure of the data collection flow to accurately describe the sample selection process and to facilitate the reader's understanding.
All patients meeting the inclusion criteria were included in the analysis, the sample size was determined by the number of patients enrolled in the study at the time of analysis. We better explain it in the result section.
- L. 108 The authors stated that the only inclusion criterion was age 18 or older, does this not include BMI or race?
BMI was indeed an inclusion criteria, as only overweight or obese patients were included in the study. All patients were caucasian.
- L. 110-112 I think the exclusion criteria should include the use of sleeping pills.
We have better specified the medications “Exclusion criteria were: 1) pregnancy and nursing; 2) diagnosed cardiovascular, neuro-logical, endocrine, and major psychiatric disorders (e.g., schizophrenia, major depression, history of psychiatric disorders); 3) diagnosis of obstructive sleep apnoea; 4) any use of medication affecting appetite, sleep and eating behaviour”.
- L. 111 There are two 3rd exclusion criteria.
Fixed it.
- L. 130-171 The description of the study items should start with the primary variable of the study. The adjustment variable, MD, should be listed after eating behaviors.
Fixed it.
- Please describe how physical activity and smoking habits were measured.
Fixed it: “Smoking and physical activity was assessed by asking participants as follows: “Do you smoke?”, “If yes, how many cigarettes per day?”, Have you ever smoked before?”, "Do you engage in any kind of structured physical activity?" and "How many hours per week do you spend doing this kind of activity?". Participants who smoke or have smoked at least 1 cigarette per day for at least 3 consecutive months were considered smokers or ex-smokers, respectively. Participants who spent 2 hours per week in any type of struc-tured exercise were considered active.”
- The authors should describe the validity of the sample size. If the sample size was insufficient to perform the analyses chosen by the authors, the robustness of the study method is estimated as low.
All patients meeting the inclusion criteria were included in the analysis, the sample size was determined by the number of patients enrolled in the study at the time of analysis. No a-priori sample size calculation was performed.
That said, the dataset contained ~94 observations per degree of freedom of the model used to describe the relationship between chronotype and either binge eating disorder or food addiction, and so should be able to adequately express the relationship without the risk of overfitting.
- L.172-185 The analysis section should be detailed so that anyone can replicate the analysis using the authors' data. The authors should report in detail on all analytical methods presented in the Results section. For example, it is difficult to discern what the dependent and independent variables in a multiple regression analysis were, respectively. It is also preferable to perform a single regression analysis. It is difficult to read that a multiple regression analysis that included the entire sample and examined the interaction between sex and chronotype and multiple regression analysis by sex, respectively, were performed. A statement regarding Figure 1 needs to be added. For the multiple regression analysis, it is necessary to specify why the forced entry method was chosen instead of the stepwise method.
We thank the reviewer for the suggestion. We performed 2 regression analysis, either having as a predictor the “binge eating score” (continuous) or the “food addiction” score (continuous) measured with the “Binge eating scale” and “Yale food addiction scale 2.0” questionnaire respectively.
Both models had the following predictor variables:
- sex (categorical: 0 = female, 1 = male)
- age (continuous, years)
- smoking (categorical: 0 = never smoked, 1 = smoker, ex smoker)
- physical activity (categorical: 0 = sedentary, 1 = physically active)
- body mass index (continuous, kg/m²)
- Mediterranean diet adherence score (continuous)
- reduced morningness-eveningness questionnaire score (continuous)
- sex × reduced morningness-eveningness questionnaire score interaction
We have more detailed the statistical analysis section.
Results
- Please rephrase these sentences in the results section so that the directionality of the results is clear throughout the results. For example, is a "slower (Eveningness)" chronotype associated with an "increase" in BES (more severe BE)?
We thank the reviewer for the suggestion. We have rephrased the sentences “We found that women showed the same prevalence of binge eating in all chrono-types (0.114, 95% CI = -0.11 – 0.34; p=0.312), while being male with overweight or obesity and a slower (eveningness) chronotype is associated with an increase in more severe binge eating behaviour (0.234, 95% CI = 0.06 – 0.41, p=0.008; -0.274, 95% CI = -0.54 - -0.01; p= 0.042), respectively). See Figure 1.”
“We found that females showed the same prevalence of food addiction in all chronotypes (0.007; 95% CI = -0.11 – 0.03; p=0.838), while being male with overweight or obesity and a slower (eveningness) chronotype is associated with an increase in more severe food ad-diction (0.234, 95% CI = 0.06 – 0.41, p=0.008; -0.274, 95% CI = -0.54 - -0.01; p= 0.042), respectively). Figure 2 shows that females tend to have higher food addiction across morn-ingness-eveningness categories, while E-type males show higher levels of food addiction.”
- L.190 The term "patients" is confusing and should be unified with "participants".
Fixed it.
- L.200 I think this sentence would be more appropriate to be moved the previous section.
Fixed it.
- If the authors think this analysis of Tables 3 and 5 are necessary, the meaning of these analyses should be described in the analysis section. The interaction between sex and chronotype is not mentioned at all in the Introduction to Methods section.
We have removed Tables 3 and 5 (original numbering).
- L.233-234 It is difficult to understand from which part of the results this sentence is derived.
While on average the model does not show a significantly higher food addiction in females vs males, the interaction term and the new Figure 2 show that females tend to have higher food addiction across morningness-eveningness categories, while morning-type males show lower levels of food addiction.
Table
- Tables and figures need to be presented in a style that can be understood independently of the text. Please revise the titles to be more specific and appropriate throughout.
Table 2. Multivariate linear regression analysis of binge eating score from the “Binge eating scale” questionnaire.
Table 3. Multivariate linear regression analysis of binge eating score from the “Binge eating scale” questionnaire.
Figure 1. Association between chronotype and binge eating, stratified by sex.
Figure 2. Association between chronotype and food addiction, stratified by sex.
- Table 1 It is unclear what the p-value represents. The authors need to show the results of the ANOVA of Table 1.
We agree with the reviewer. We added the following specifications:
“° p<0,05 vs M-types; * p<0,001 vs M-types.
rMEQ scores were significantly lower in E-Types compared to M-Types (10±1 vs 19±1, p < 0,001), as expected, and Mediterranean Diet adherence was lower in E-Types (6±2 vs 7±2, p < 0.05)”.
- Table 2 What does "Estimate" mean? From the values, it looks like a beta. Additionally, only the Estimate for Sex has a very different value. Please correct it if necessary.
We agree with the reviewer. We have replaced “estimate” with beta and corrected the Sex values.
- I am not sure what the difference is between Table 2 and Table 4. They have almost the same title, but Table 4 may be the result of FA.
FIxed it.
Discussion
- L.239-246 The authors should also mention the results of Figure 1 in this paragraph.
We have changed the Figure 1.
- L.247 What "these results" mean ? If all the results are consistent with previous studies, the novelty of this study is lost.
We have better explained the differences between our study and others, such as the different population (not healthy but with overweight and obesity, not children or adolescents but adults, both male and females) and the combination of both sex and lifestyle in studying the association between chronotype and eating behaviour. See Discussion section.
- L. 312-313 It is unclear from what analysis the authors report that the sample size of men was lacking. Moreover, if the sample size was indeed lacking, the academic value of this study would be greatly compromised.
We agree with the reviewer.
Our comment referred to the disproportion between men and women and not to the small sample size. As can be seen from the degrees of freedom of the model (see response 14), the sample size is sufficient to avoid bias in the interpretation of the results. We better explained: “The sample size of men was lacking compared to women; however, the dataset contained ~94 observations per degree of freedom of the model used to describe the relationship be-tween chronotype and either binge eating disorder or food addiction, and so should be able to adequately express the relationship without the risk of overfitting of bias in the in-terpretation in the results”.

Reviewer 2 Report
In this work the authors aimed to study the relationship between the chronotype and the risk of developing eating disorders. The authors recruited 750 subjects and determined through a series of questionnaires the diet, the chronotypes, and the eating behavior. The aim of the study is very interesting. With regards to personalized medicine approaches, the understanding of circadian rhythm and how different chronotypes respond to treatment is highly relevant. However, the current study is limited in its conception, and its conclusion difficult to interpret.
Overall, the study is quite hard to read. The overuse of acronyms (FA, BE, rMEQ, YFAS, MD, MEDAS, BES) makes some sentences barely readable.This is one example “the MD was found protective only in women (-0.136, 95% CI =-0.26 227 - -0.02, p=0.043), we analyzed the relationship between chronotype (rMEQ score) and FA at different level of MD (MEDAS)”. I think that some effort can be made to improve the readability of the study.
It is sometimes difficult to understand how all these parameters are correlated to each other. Overall the authors do not find association between the chronotype and food disorder. They found that only men with an evening chronotype have a tendency to report food addiction. Men are under-represented in this study (30%) and maybe the lower representation may create a bias. Overall the all study is only correlation and there is no data that could discriminate between a cause, a consequence or just a correlation. How can we conclude that “ being male and E-Type increase risk of BE and/or FA prevalence”. What is the cause or the consequence?. Do these E-type have chronic circadian disruption?
The representation of the data could also be improved. The authors show different tables with association of the chronotype and either the Food addiction, Binge Eating Scale etc.. but the tables do not have different chronotypes? How has figure 1 been generated?
So part of the study are difficult to read.
Author Response
#Reviewer 2
In this work the authors aimed to study the relationship between the chronotype and the risk of developing eating disorders. The authors recruited 750 subjects and determined through a series of questionnaires the diet, the chronotypes, and the eating behavior. The aim of the study is very interesting. With regards to personalized medicine approaches, the understanding of circadian rhythm and how different chronotypes respond to treatment is highly relevant. However, the current study is limited in its conception, and its conclusion difficult to interpret.
We thank the reviewer for the opportunity to significantly improve our work.
Overall, the study is quite hard to read. The overuse of acronyms (FA, BE, rMEQ, YFAS, MD, MEDAS, BES) makes some sentences barely readable.This is one example “the MD was found protective only in women (-0.136, 95% CI =-0.26 227 - -0.02, p=0.043), we analyzed the relationship between chronotype (rMEQ score) and FA at different level of MD (MEDAS)”. I think that some effort can be made to improve the readability of the study.
We have significantly revised all English and drastically reduced the use of abbreviations throughout the paper.
It is sometimes difficult to understand how all these parameters are correlated to each other. Overall the authors do not find association between the chronotype and food disorder.
We aimed to describe the relationship between chronotype and either binge eating disorder or food addiction, adjusting for known factors influencing the risk of developing binge eating or food addiction (age,BMI,adherence to mediterranean diet, physical activity level), allowing for a different relationship between sexes, as sexes tend to behave differently in the pathogenesis of these disorders. So we have retained Table 2 and 4 and eliminated table 3 and 5 from the paper and better explained these aspects in the introduction and discussion section.
We did not find direct association between the chronotype and the food disorders’ score, but because the relationship is mediated by the sex (see the significant interaction between sex and chronotype with binge eating and food addiction scores (-0.406; 95% CI = -0.79 - -0.002; p= 0.037); -0.244, 95% CI = -0.18 – -0.31, p= 0.039, respectively).
They found that only men with an evening chronotype have a tendency to report food addiction. Men are under-represented in this study (30%) and maybe the lower representation may create a bias.
As widely known in literature, fewer men than women spontaneously refer to a specialised dietary intervention centre. At the suggestion of both reviewers, however, we checked and the sample size is sufficient to avoid bias in the interpretation of the results.
All patients meeting the inclusion criteria were included in the analysis, the sample size was determined by the number of patients enrolled in the study at the time of analysis. No a-priori sample size calculation was performed.
That said, the dataset contained ~94 observations per degree of freedom of the model used to describe the relationship between chronotype and either binge eating disorder or food addiction, and so should be able to adequately express the relationship without the risk of overfitting.
Thanks to the reviewers’ suggestion, we improved the statistical analysis section and the Discussion to better describe these aspects.
Overall the all study is only correlation and there is no data that could discriminate between a cause, a consequence or just a correlation. How can we conclude that “ being male and E-Type increase risk of BE and/or FA prevalence”. What is the cause or the consequence?. Do these E-type have chronic circadian disruption?
We agree with the reviewer. We rephrased the results’ section and we better describe the association and not the risk, underlining in the Discussion section the limit of the cross sectional design.
“We found that women showed the same prevalence of binge eating in all chrono-types (0.114, 95% CI = -0.11 – 0.34; p=0.312), while being male with overweight or obesity and a slower (eveningness) chronotype is associated with an increase in more severe binge eating behaviour (0.234, 95% CI = 0.06 – 0.41, p=0.008; -0.274, 95% CI = -0.54 - -0.01; p= 0.042), respectively). See Figure 1.”
“We found that females showed the same prevalence of food addiction in all chronotypes (0.007; 95% CI = -0.11 – 0.03; p=0.838), while being male with overweight or obesity and a slower (eveningness) chronotype is associated with an increase in more severe food ad-diction (0.234, 95% CI = 0.06 – 0.41, p=0.008; -0.274, 95% CI = -0.54 - -0.01; p= 0.042), respectively). Figure 2 shows that females tend to have higher food addiction across morn-ingness-eveningness categories, while E-type males show higher levels of food addiction.”
“the cross-sectional design makes it impossible to establish a cause-and-effect relationship”.
We also improved the explanation of the possible role of the circadian disruption typical of E-types in the link between chronotype and food addiction/binge eating.
“Little but growing evidence is trying to investigate how circadian rhythm disruptions may characterize either symptoms or modulators of overeating, which could be explained by the link between insomnia and night eating syndrome [34], but also by the loss of the cir-cadian rhythmicity of metabolic and hedonic regulation of feeding due to the social jet-lag and a gradual increase of prevalence of emotional eating [23,27]. These results are also consistent with the findings of Aoun et al [24]: they found that healthy male M-types have lower uncontrolled eating and more cognitive restraint related to food when compared to male E-types, which are more likely to engage in addictive eating patterns [12]”.
The representation of the data could also be improved. The authors show different tables with association of the chronotype and either the Food addiction, Binge Eating Scale etc.. but the tables do not have different chronotypes? How has figure 1 been generated?
We agree with the reviewer. We have deleted the tables 3 and 5 (original number) and the figure 1 and we added other two figures that better explain the two models on the association between chronotype and binge eating/food addiction, stratified by sex.

Round 2
Reviewer 1 Report
I think the manuscript has been significantly improved. The revised version makes the arguments clearer and emphasizes the value of the study more than the original version. As a result, this raises the question of whether the key word in this paper is "sex and chronotype interaction" rather than sex differences per se. If my reasoning is correct, please revise the authors’ wording, from the title to the discussion, to take this into account and to properly convey the focus of the study. Is there any mention in prior research of a gender and chronotype interaction? It was not mentioned in the original version whether the result that only the interaction, not the main effect, was significant is consistent with prior studies or not, or whether it was not considered in the previous studies. The value of the study could be further enhanced by revising the novelty of the study, especially in the introduction and the discussion based on the focus of the study (interactions). Please revise the description to be consistent from the title/introduction to the discussion based on the focus of the authors' study.
The following several comments will help you reorganize this manuscript.
Revisions to the first revision
Title.
Thanks for the revision. I think it would be easier to convey the significance of this study if the interaction between gender and chronotype is fully described.
Method
8. Insufficient revision. An explanation needs to be added so that potential readers can understand the sample selection process. It remains unclear how many people were recruited, how many responded (n=750?), how many were included in the analysis (n=750?), and what the eligibility rate was.
14. It is recommended that a power analysis be used to determine the sample size before conducting the survey. The authors' claim concerning sample size validity is not evidence-based and lack sufficient reference. Since the results of the gender-specific analyses were removed, it is no longer necessary to demonstrate the validity of the sample size for females/male, respectively. Please cite the literature for evidence of validity using degrees of freedom. Even if the sample size was not estimated before the study was conducted, the authors need to explain in the text the validity of the results of the analysis based on the evidence: a post hoc test should be conducted to describe the effect size using the sample size, power (1-β), and significance level (α).
15. The linear regression analysis was revised for clarity. However, there is still a lack of explanation regarding the use of the forced entry method in the multiple regression analysis (and preferably the reason for this). Please add this to the table annotations.
Additional comments on the revised section
2-1. In order to examine interactions in multiple regression analysis, multicollinearity must be considered.
2-2. L.243-247, 257-261 Are the results of the tables removed from the original version still available? It is unclear from which part of the tables and figures are taken. The authors should describe what can be read from the results in Figure 1, just as the authors did in L. 261-263 for FA.
2-3. l. 261-263 I did not understand the meaning of this sentence.
2-4. The discussion needs to be substantially restructured based on the revisions. In particular, please restructure this section to focus on the fact that the direct effect was non-significant and the interaction was significant.
2-5. l.270-276 What this statement indicates has been removed in the revised version.
2-6. L.277 LATER chronotype instead of LOWER
2-7. l.277-280 In the revised version, only the interaction was significant, which was not consistent with the results of previous studies. Throughout the section, please review whether the authors' results in the revised version are reflected in the discussion.
Author Response
#Reviewer 1 - round 2
I think the manuscript has been significantly improved. The revised version makes the arguments clearer and emphasizes the value of the study more than the original version. As a result, this raises the question of whether the key word in this paper is "sex and chronotype interaction" rather than sex differences per se. If my reasoning is correct, please revise the authors’ wording, from the title to the discussion, to take this into account and to properly convey the focus of the study. Is there any mention in prior research of a gender and chronotype interaction? It was not mentioned in the original version whether the result that only the interaction, not the main effect, was significant is consistent with prior studies or not, or whether it was not considered in the previous studies. The value of the study could be further enhanced by revising the novelty of the study, especially in the introduction and the discussion based on the focus of the study (interactions). Please revise the description to be consistent from the title/introduction to the discussion based on the focus of the authors' study.
We thank the reviewer for the suggestion. We added the following sentences in the introduction:
“Sex differences in chronotyping are also reported, with most studies showing that males tend to be more likely than females to be in the evening/late chronotypes”
“Despite these sex differences in both eating behaviour and chronotype, to the best of our current knowledge, no previous study has investigated whether and how the interac-tion between sex and chronotype may influence the development of various pathologies as alterations in eating behavior. In such a scenario, we carried out a cross-sectional study to investigate the sex dif-ferences in the relationship between chronotype and either binge eating disorder or food addiction in Caucasian adults with overweight or obesity, considering lifestyle factors.”
Revisions to the first revision
Title.
Thanks for the revision. I think it would be easier to convey the significance of this study if the interaction between gender and chronotype is fully described.
We thank the reviewer for the suggestion. We have further modified the title in “Sex differences in the relationship between chronotype and eating behavior: a focus on binge eating and food addiction”
Method
- Insufficient revision. An explanation needs to be added so that potential readers can understand the sample selection process. It remains unclear how many people were recruited, how many responded (n=750?), how many were included in the analysis (n=750?), and what the eligibility rate was.
We added Figure 1 (participant flow diagram) and the following sentence in the Results section: “Of the 856 participants who came to ICANS-DIS, 106 were ecluded: 88 (10%) did not meet the inclusion criteria and 18 (2%) did not complete all questionnaires. 750 participants were included in the analysis with an eligibility rate of 88% (see Figure 1).”
- It is recommended that a power analysis be used to determine the sample size before conducting the survey. The authors' claim concerning sample size validity is not evidence-based and lack sufficient reference. Since the results of the gender-specific analyses were removed, it is no longer necessary to demonstrate the validity of the sample size for females/male, respectively. Please cite the literature for evidence of validity using degrees of freedom.
Fixed it:
- Harrell, Frank E. Regression modeling strategies: with applications to linear models, logistic regression, and survival analysis. Vol. 608. New York: Springer, 2001
Even if the sample size was not estimated before the study was conducted, the authors need to explain in the text the validity of the results of the analysis based on the evidence: a post hoc test should be conducted to describe the effect size using the sample size, power (1-β), and significance level (α).
We agree with the reviewer. We estimated the sample size according to pmsampsize(type = "c", rsquared =, parameters = 8, intercept =, sd = ) https://www.bmj.com/content/368/bmj.m441 and based on previous and similar studies investigating the predictors of binge eating or food addiction
- For BES based on https://doi.org/10.1108/NFS-02-2021-0062: Minimum sample size required for the development of a new model based on user input = 243
- For YFAS based on https://doi.org/10.3389%2Ffpsyt.2023.1200021: Minimum sample size required for the development of a new model based on user input = 571
We added this explanation in the Statistical analysis section.
- The linear regression analysis was revised for clarity. However, there is still a lack of explanation regarding the use of the forced entry method in the multiple regression analysis (and preferably the reason for this). Please add this to the table annotations.
Fixed it. We added the following sentence:
“The variables were selected within the known predictor of either binge eating or food addiction that would explain part of the variance of these outcomes, producing more precise estimate of the effect of chronotype (dependent on sex) on the outcomes”
Additional comments on the revised section
2-1. In order to examine interactions in multiple regression analysis, multicollinearity must be considered.
We thank the reviewer for the suggestion. We added the following part in the methods section: “To exclude multicollinearity, we conduct a test with Variance Inflation Factors (VIFs). As the reviewer can see from the tables below, all VIFs are significantly below 5 to 10 and none of these variables are above the average threshold. So, we kept all of them, because it's usually bad to remove variables unless absolutely necessary (Kim JH. Multicollinearity and misleading statistical results. Korean J Anesthesiol. 2019 Dec;72(6):558-569. doi: 10.4097/kja.19087)”
Here the results of the multicollinearity test:
Model |
Non-standard coefficients |
Standardised coefficients |
t |
Sign. |
Collinearity |
|||
B |
Standard error |
Beta |
Tolerance |
VIF |
||||
1 |
(Constant) |
11,746 |
2,208 |
|
5,321 |
0,000 |
|
|
age |
-0,112 |
0,025 |
-0,172 |
-4,545 |
0,000 |
0,949 |
1,053 |
|
sex |
-2,373 |
0,590 |
-0,151 |
-4,019 |
0,000 |
0,961 |
1,040 |
|
smoking |
0,345 |
0,401 |
0,032 |
0,860 |
0,390 |
0,985 |
1,015 |
|
physical activity |
-0,991 |
0,637 |
-0,058 |
-1,556 |
0,120 |
0,968 |
1,033 |
|
bmi |
0,238 |
0,035 |
0,254 |
6,709 |
0,000 |
0,948 |
1,055 |
|
med_score |
-0,110 |
0,157 |
-0,027 |
-0,699 |
0,485 |
0,931 |
1,075 |
|
meq_score |
-0,143 |
0,082 |
-0,066 |
-1,734 |
0,083 |
0,953 |
1,049 |
|
a. Dependent variable: bes |
Model |
Non-standard coefficients |
Standardised coefficients |
t |
Sign. |
Collinearity |
|||
B |
Standard error |
Beta |
Tolerance |
VIF |
||||
1 |
(Constant) |
0,765 |
0,826 |
|
0,926 |
0,355 |
|
|
age |
-0,003 |
0,009 |
-0,014 |
-0,374 |
0,709 |
0,949 |
1,053 |
|
sex |
-0,803 |
0,221 |
-0,136 |
-3,636 |
0,000 |
0,961 |
1,040 |
|
smoking |
-0,065 |
0,150 |
-0,016 |
-0,435 |
0,664 |
0,985 |
1,015 |
|
physical activity |
-0,084 |
0,238 |
-0,013 |
-0,354 |
0,723 |
0,968 |
1,033 |
|
bmi |
0,113 |
0,013 |
0,321 |
8,510 |
0,000 |
0,948 |
1,055 |
|
med_score |
-0,128 |
0,059 |
-0,083 |
-2,171 |
0,030 |
0,931 |
1,075 |
|
meq_score |
-0,041 |
0,031 |
-0,050 |
-1,330 |
0,184 |
0,953 |
1,049 |
|
a. Dependent variable: yfas_score |
2-2. L.243-247, 257-261 Are the results of the tables removed from the original version still available? It is unclear from which part of the tables and figures are taken. The authors should describe what can be read from the results in Figure 1, just as the authors did in L. 261-263 for FA.
We changed the sentence in: “Figure 2 shows that females tend to have higher binge eating than males regardless of morningness-eveningness categories, while E-type males show higher levels of binge eating than M-type males, with levels similar to females”.
2-3. l. 261-263 I did not understand the meaning of this sentence.
We changed the sentence in: “Figure 3 shows that females tend to have higher food addiction than males regardless of morningness-eveningness categories, while E-type males show higher levels of food addiction than M-type males, with levels similar to females”.
2-4. The discussion needs to be substantially restructured based on the revisions. In particular, please restructure this section to focus on the fact that the direct effect was non-significant and the interaction was significant.
We thank the reviewer for the suggestion. We added the following paragraph in the discussion section and we rephrased other sentences to underline the novelty of the interaction:
“In particular, we found that the direct effect of chronotype on binge eating and food addiction scores was not significant, as reported in previous studies, but the interaction between sex and chronotype was significant, highlighting how the effect of chronotype on the development of disorders such as eating disorders is mediated by sex differences. This may be due to the fact that our sample consists of middle-aged participants (30-65 years), in whom sex differences in chronotype expression are still relevant, but also to the fact that many aspects of lifestyle, such as smoking, exercise and dietary patterns, were taken into account when examining this relationship. Future studies would need to be evaluated in the older population, where these differences tend to diminish until they disappear altogether.”
2-5. l.270-276 What this statement indicates has been removed in the revised version.
Fixed it. See 2.2
2-6. L.277 LATER chronotype instead of LOWER
Fixed it.
2-7. l.277-280 In the revised version, only the interaction was significant, which was not consistent with the results of previous studies. Throughout the section, please review whether the authors' results in the revised version are reflected in the discussion.
We changed the sentence in : “E-types males show higher levels of both binge eating and food addiction than M-types males, and reach levels similar to those of females, which are higher in all categories of chronotype.. In fact, we found no correlations between chronotype and pathological overeating in women females at any level, confirming only young age, high BMI and low adherence to Mediterranean Diet as predictors of food addiction”.
